# Efficacy of Alkaline Phosphatase in Critically Ill Patients with COVID-19: A Multicentre Investigator-Initiated Double-Blind Randomised Placebo-Controlled Trial

**DOI:** 10.3390/biomedicines12040723

**Published:** 2024-03-25

**Authors:** Anouk Pijpe, Stephan G. Papendorp, Joost W. van der Heijden, Ben Vermin, Iris Ertugrul, Michael W. J. Ritt, Björn Stessel, Ina Callebaut, Albertus Beishuizen, Marcel Vlig, Joost Jimmink, Henk J. Huijgen, Paul P. M. van Zuijlen, Esther Middelkoop, Evelien de Jong

**Affiliations:** 1Department of Intensive Care, Red Cross Hospital, Vondellaan 13, 1942 LE Beverwijk, The Netherlands; apijpe@rkz.nl (A.P.); spapendorp@rkz.nl (S.G.P.);; 2Department of Plastic, Reconstructive and Hand Surgery, Amsterdam UMC Location Vrije Universiteit, Boelelaan 1117, 1081 HV Amsterdam, The Netherlands; pvanzuijlen@rkz.nl (P.P.M.v.Z.); e.middelkoop@amsterdamumc.nl (E.M.); 3Amsterdam Movement Sciences, Tissue Function and Regeneration, Boelelaan 1117, 1081 HV Amsterdam, The Netherlands; 4Association of Dutch Burn Centres, Zeestraat 27-29, 1941 AJ Beverwijk, The Netherlands; mvlig@burns.nl; 5Burn Centre, Red Cross Hospital, Vondellaan 13, 1942 LE Beverwijk, The Netherlands; jjimmink@rkz.nl; 6Department of Internal Medicine, Spaarne Gasthuis, Spaarnepoort 1, 2134 TM Hoofddorp, The Netherlands; jvanderheijden@spaarnegasthuis.nl; 7Department of Intensive Care Medicine, Spaarne Gasthuis, Spaarnepoort 1, 2134 TM Hoofddorp, The Netherlands; bvermin@spaarnegasthuis.nl (B.V.);; 8Department of Intensive Care Medicine, Jessa Hospital, Stadsomvaart 11, 3500 Hasselt, Belgium; bjorn.stessel@jessazh.be (B.S.); ina.callebaut@jessazh.be (I.C.); 9LCRC, Faculty of Medicine and Life Sciences, University Hasselt, Agoralaan, 3590 Diepenbeek, Belgium; 10Intensive Care Center, Medisch Spectrum Twente, Koningsplein 1, 7512 KZ Enschede, The Netherlands; b.beishuizen@mst.nl; 11Department of Clinical Chemistry, Red Cross Hospital, 1942 LE Beverwijk, The Netherlands; hhuijgen@rkz.nl; 12Department of Plastic Reconstructive and Hand Surgery, Red Cross Hospital, Vondellaan 13, 1942 LE Beverwijk, The Netherlands; 13Emma Children’s Hospital, Pediatric Surgical Center, Amsterdam UMC Location Academic Medical Center, Meibergdreef 9, 1105 AZ Amsterdam, The Netherlands

**Keywords:** alkaline phosphatase, COVID-19, mechanical ventilation, inflammatory response

## Abstract

Background: Efforts to identify therapies to treat hospitalised patients with COVID-19 are being continued. Alkaline phosphatase (AP) dephosphorylates pro-inflammatory adenosine triphosphate (ATP) into anti-inflammatory adenosine. Methods: In a randomised controlled trial, we investigated the safety and efficacy of AP in patients with SARS-CoV-2 infection admitted to the ICU. AP or a placebo was administered for four days following admission to the ICU. The primary outcome was the duration of mechanical ventilation. Mortality in 28 days, acute kidney injury, need for reintubation, safety, and inflammatory markers relevant to the described high cytokine release associated with SARS-CoV-2 infection were the secondary outcomes. Results: Between December 2020 and March 2022, 97 patients (of the intended 132) were included, of which 51 were randomised to AP. The trial was terminated prematurely based on meeting the threshold for futility. Compared to the placebo, AP did not affect the duration of mechanical ventilation (9.0 days vs. 9.3 days, *p* = 1.0). No safety issues were observed. After 28 days, mortality was 9 (18%) in the AP group versus 6 (13%) in the placebo group (*p* = 0.531). Additionally, no statistically significant differences between the AP and the placebo were observed for the other secondary outcomes. Conclusions: Alkaline phosphatase (AP) therapy in COVID-19 ICU patients showed no significant benefits in this trial.

## 1. Introduction

COVID-19, caused by severe acute respiratory syndrome coronavirus 2 (SARS-CoV-2), led to an unprecedented global burden, and simultaneously, it has acted as a catalyst for new medical therapies and insights. While vaccination remains the most effective strategy to curb the spread and severity of SARS-CoV-2, ongoing efforts to identify therapies to prevent further clinical deterioration in infected individuals continue [1]. The most severe stage of COVID-19 infection may result in acute respiratory distress syndrome (ARDS), as well as in extra-pulmonary systemic inflammation, such as acute kidney injury due to cytokine release and activation of the complement system [2]. This excessive immune response was already recognised early during the pandemic as a potential therapeutic target [3]. Several immunomodulatory therapies have since been studied [4]. Nonspecific approaches, such as corticosteroid therapy, and more specific approaches, such as interleukin (IL) 1 and 6 inhibitors, have shown modest disease attenuation [5].

In the context of inflammatory responses, adenosine triphosphate (ATP), released by inflammatory cells, contributes to the propagation of inflammation and subsequent tissue damage. In addition to apoptosis and pyroptosis, uncontrolled cell death can occur, which leads to the passive release of ATP into the extracellular space. This unregulated ATP release serves as an immunological danger signal, activating purinergic receptors and initiating inflammatory cascades [6,7].

In COVID-19 infection, the hyperinflammatory response is thought to be (partially) driven by the outpour of ATP from pyroptotic cells. Alkaline phosphatase (AP) has been reported to exert anti-inflammatory effects by acting on these pro-inflammatory extracellular nucleosides and converting them into anti-inflammatory nucleosides through dephosphorylation [8]. Furthermore, preclinical and clinical models have demonstrated that AP is capable of restoring the integrity of various epithelial barriers, including the gut intestinal epithelium, the renal tubular epithelium, and the blood–brain barrier [9,10,11,12,13,14,15]. It is, therefore, hypothesised that AP may also restore the integrity of alveoli in the lungs [11]. Hence, we hypothesise that AP may be a safe and efficacious agent in reducing ATP, and thus, inflammation in COVID-19 infection, maintaining alveolar integrity and potentially leading to a reduction in the need for mechanical ventilation and a shorter duration of mechanical ventilation In intubated patients.

## 2. Materials and Methods

### 2.1. Trial Design

This trial was conducted in accordance with the protocol and consensus ethical principles of international guidelines, including the Declaration of Helsinki and the Good Clinical Practice Guidelines (GCP). The protocol and amendments were reviewed and approved by the Institutional Ethics Committee (IEC) in the relevant centres prior to starting the trial. This study was designed as a multicentre investigator-initiated double-blind randomised placebo-controlled trial investigating the safety and efficacy of AP in patients with COVID-19 infection admitted to the intensive care unit. An independent safety monitoring board (DSMB) supervised the trial and was not involved in the design or conduct of the trial, nor in the final statistical analyses. The DSMB convened after the enrolment of 20, 40, and 80 patients and again after 97 patients. After the last interim analysis, the DSMB judged that the a priori stopping threshold for futility had been met and recommended an early termination of the study. The study was open for inclusion between December 2020 and March 2022.

### 2.2. Participants

Eligible participants were patients ≥18 years of age with a confirmed or suspected SARS-CoV-2 infection who were admitted to the ICU and required airway support with a SpO_2_ < 90% or PaO_2_/FiO_2_ < 200 mmHg. The confirmation of COVID-19 infection (by PCR and/or specific antigen test) was required during admission. Patients who were over 80 years of age; pregnant or lactating; expected to have fatal disease within 24 h; had a known history of dialysis (renal replacement therapy, RRT) or a decision had been made to initiate RRT within 24 h after the planned start of the. study drug administration; known to have advanced chronic liver disease as confirmed by a Child–Pugh C; a known history of immune system impairment by disease, such as patients with HIV and with a CD4 count of less than 200 cells/mm; and neutropenic patients (<0.5 × 10^9^/L) were excluded. As the purpose of this study was to demonstrate the efficacy of AP in a real-life setting, patients on corticosteroid therapy were not excluded from the study. The use of systemic corticosteroids was monitored and recorded throughout the trial, including dexamethasone 6 mg once daily, used as routine treatment for severe COVID-19, which was not an exclusion criterion. Also, patients receiving one dose of tocilizumab on top of the standard of care for COVID-19 were also not excluded from this trial.

### 2.3. Sample Size

At the time of protocol development, the mean number of days COVID-19 patients were on mechanical ventilation was 10 days (ICNARC Case mix program database 17 April 2020). With AP, we aimed to reduce the days on mechanical ventilation by 15%. With 80% power, a 2-sided alpha of 0.05, and a standard deviation of 3, we needed to include at least 63 evaluable patients to prove the benefit of AP for the primary endpoint in each arm (AP and placebo), resulting in a total sample size of 126 patients. With an estimated drop-out of 5% (e.g., withdrawal of informed consent), 132 patients needed to be included.

### 2.4. Randomisation and Treatment

The investigational medicinal product (IMP) was RESCAP^®^ (RESCuing Alkaline Phosphatase; Alloksys Life Sciences, Wageningen, The Netherlands). The patients were randomly allocated 1:1 to receive either AP or a saline placebo using a centralised block randomisation procedure (blocks of 20 patients) with a computer-generated list produced by an independent contract research organisation, Aix Scientifics^®^ (Aachen, Germany). The patients received a bolus of 1.000 IU, followed by 10.000 IU a day of AP or a bolus of the placebo and infusion of the placebo for four days. The chosen dose regime was retrieved from modelling studies of kinetics data from two clinical studies with the same AP and the same total dose of 10,000 units, of which 1000 units were given as a bolus [16]. The treatments were blinded to the participants, all healthcare workers, including the investigators recruiting and allocating participants, and the investigators analysing the data. Unblinding took place following the completion of the statistical analyses.

### 2.5. Outcomes

The primary outcome was the duration of mechanical ventilation (days) and the number of patients in need of mechanical ventilation (%). Secondary outcomes included mortality in 28 days, safety, daily levels of routine clinical lab parameters, (whose AF was blinded until conclusion of the study), and daily levels of the following inflammatory markers: IL-2, IL-4, IL-6, IL-10, IP10 (CXCL10), IL1b, TNF-a, MCP-1 (CCL2), IL17A, IFN-y, IL-12p70, and free active TGFb.

### 2.6. Study Procedures and Sample Analyses

All patients who met the inclusion criteria were asked to participate. For patients who could not give consent themselves, a legal representative was asked for consent. If the legal representative was not allowed or able to come to the hospital, verbal consent could be given by phone or in an online meeting, and written consent would follow within the next three days. After consent, the IMP was started as soon as possible, to potentially gain the most benefit from the intervention.

During the study period, for the analysis of the inflammatory markers, in the Dutch hospitals, an extra tube (10 mL) of whole blood was collected simultaneously with the routine clinical blood collection: once daily during ICU admission until the day of discharge or day 14, whichever came first. The local clinical chemistry laboratories centrifuged the blood and separated plasma was locally stored in 3 aliquots with an minimum of 1.0 mL at −80 °C. At the end of the entire study, all samples were transported, under controlled conditions, to the preclinical research lab of the Association of Dutch Burn Centres, Beverwijk, The Netherlands. Inflammatory markers were analysed using the Human Essential Immune Response Panel (Biolegend Europe, Amsterdam, The Netherlands) according to the manufacturer’s protocol. The samples were measured on a MACSQuant10 (Miltenyi Biotec, Leiden, The Netherlands), and the data were processed using FlowLogic 8.7 (Inivai Technologies, Melbourne, VIC, Australia).

During the pandemic, patient transfers occurred frequently. When a patient was transferred to another hospital during the study period, efforts were undertaken to at least complete the post-discharge follow-up regarding clinical outcomes.

### 2.7. Statistical Analyses

The statistical analyses were performed blinded, meaning the patients were allocated to either treatment A or B and analysed as such. Descriptive statistics were used to report the data on the baseline characteristics and clinical outcomes. Depending on the data type and distribution, statistically significant differences between the treatment arms were determined by independent *T*-tests, the Wilcoxon signed-rank test, the chi-square test or Fisher’s exact test with an observed count < 10 or an expected count < 5. Survival was also investigated by the Kaplan–Meier method. The effect of the treatment on the course of routine clinical lab parameters and inflammatory markers was investigated by a linear mixed model, based on the daily lab data up to day 14, with a random intercept of ID and a variance component (VC) covariance structure. Centre and time dependence were investigated by testing the addition of a random intercept of the centre and a random slope of the intervention. In the case that groups A and B differed in one or more baseline characteristics, we investigated possible confounding by multivariate analysis. Potential effect modification was also investigated and adjusted for, if necessary (e.g., based on treatment with tocilizumab). The overall fit of the models was tested by comparing the −2 log likelihood. Potential short-term effects were investigated by analyses of the data up to day 7. All analyses were performed by both the intention to treat (ITT, *N* = 97) and per protocol (PP, *N* = 72) principles. Statistical significance was set at *p* < 0.05. Data processing and statistical analysis were performed using R studio version 4.3.3 and SPSS 25.

## 3. Results

Between December 2020 and March 2022, a total of 301 patients were screened, of which 204 patients were excluded for not meeting the inclusion criteria (111), decline to participate (37) or other reasons (56) (Figure 1).

A total of 97 patients were randomised, of which 51 patients were assigned to the AP group and 46 patients to the placebo group. The demographics and baseline characteristics of the patients were similar in the two treatment arms, except for history of respiratory disease, which was more frequent in the AP group (Table 1).

The median age of the patients in the trial was 61 years (IQR 54–69). The majority (71%) were male. All patients tested positive for SARS-CoV-2 by qualitative PCR at randomisation. About one-third (32%) of the patients were treated with tocilizumab, and 74% received antibiotics for a median of 5 days during the study.

On 31 March 2022, trial enrolment was halted. After the third interim, the DSMB ordered another interim analysis of the primary outcome of the 97 patients enrolled. After interpreting the blinded results, the DSMB determined that the a priori stopping threshold for futility had been met and recommended early termination of the study. Therefore, the trial was terminated prematurely after the enrolment of 97 patients.

In total, 25 (26%) patients were transferred to another hospital or ward during the study period, of whom 9 (9%) were lost to follow-up, i.e., no information regarding their outcomes could be assessed. There was no difference in follow-up completeness by treatment arm.

### 3.1. Primary Outcome

In the AP group, 35 patients (69%) required mechanical ventilation, compared to 26 patients (57%) in the placebo group (*p* = 0.218) (Table 2). Among the patients who required mechanical ventilation, the duration of mechanical ventilation was similar in both treatment arms (median of 9.0 days versus 9.3 days; *p* = 1.000). There was no difference in the results of the ITT and PP analyses.

### 3.2. Secondary Outcomes

The overall mortality rate at 28 days was 15.4%, and no difference was observed between the treatment arms (*p* = 0.531) (Table 2 and Figure 2). The mortality <14 days did not differ either (*p* = 0.226). The most frequent cause of death was progressive COVID-19 pneumonia (66%). Two patients died due to complications of COVID-19-associated pulmonary aspergillosis, one in the intervention group and one in the placebo group. Additionally, three patients died from severe superinfection, with two in the intervention group and one in the placebo group. The length of ICU stay and length of hospital stay were not different between the treatment arms.

No other serious adverse events (SAEs) were observed. All adverse events, which were abnormal routine clinical laboratory values or the development of pulmonary embolism, were reviewed and determined to be part of the normal clinical course of severe COVID-19 infection and not related to the IMP/placebo. Among the study participants, two individuals in the placebo group and five in the intervention group developed stage 3 acute kidney injury (AKI) according to the KDIGO criteria (Kidney Disease Improving Global Outcomes). One patient in the AP group developed a rash on day one of infusion; however, antibiotics were administered simultaneously. No signs of anaphylaxis were seen in this patient. 

The course of levels of AP per treatment group, which were made available by the clinical laboratory after closure of the study, are depicted in Figure 3. 

Longitudinal analyses of the levels of the routine clinical laboratory values of the PaO_2_/Fio_2_ ratio, CRP, haemoglobin, leukocyte count, and creatinine revealed no differences between the treatment arms (Table 3).

The results were not different for the short-term effect analysis of the data up to day 7. Regarding the additional inflammatory markers, no statistically significant differences between the AP and placebo were observed (Table 4).

Moreover, there was no significant interaction between intervention and time for any of the parameters. Visual inspection of the data revealed a suggestive but statistically non-significant trend concerning the levels of IL6, IL10, IP10, and MCP–1 (see Figure 4a–c). The results were not different for the short-term effect analysis of the data up to day 7.

There were no differences between the ITT and PP analyses for any of the secondary outcome measures.

### 3.3. Subgroup Analyses

Subgroup analyses were conducted to explore potential differences, specifically focusing on patients with initially elevated IL-6 levels at inclusion and those who did not receive tocilizumab. This revealed no discernible distinctions in the outcomes among these subgroups.

## 4. Discussion

This trial is the first study examining the efficacy and safety of AP in severely ill SARS-CoV-2 patients. No safety issues were observed in this population with the use of AP. The patients treated with AP did not show a shorter duration of mechanical ventilation compared to those who received the placebo. There was no difference in mortality between the patients who received AP and those who received the placebo. Additionally, the administration of AP did not affect the secondary outcome measures, such as the duration of hospital stay and the duration of ICU admission. Furthermore, no significant differences were observed in the infection parameters or levels of inflammatory markers.

The concept of immunomodulatory therapy was adapted early during the COVID-19 pandemic. While immunosuppressive drugs have the potential to dampen an excessive immune response, there is a risk of reducing the necessary antiviral immune response. This study was initiated in the early stages of the pandemic before routine treatments, such as corticosteroids, and later, monoclonal antibodies, specifically IL-6 inhibitors, were introduced. Hypothetically, AP could offer specific advantages due to its short half-life and its potential to intervene in the onset of the cytokine cascade.

Although our findings regarding a potential benefit of AP concerning cytokine levels were statistically not significant, nor did they translate into differences in the clinical outcome parameters, we believe this strategy merits further investigation for several reasons. First, in the context of severe COVID-19 infections in the intensive care setting, particular certain cytokines have emerged as crucial determinants of disease progression. Modulating these cytokine levels or their downstream effects may be a strategy to restore a dysregulated immune response. Interleukin-6 (IL-6), interferon-gamma-inducible protein 10 (IP-10), and monocyte chemoattractant protein-1 (MCP-1) showed downward trends after treatment with RESCAP. Notably, IL-6 has gained significant attention due to its pivotal role in the inflammatory response. Elevated levels of IL-6 have been associated with severe respiratory distress and cytokine storm, contributing to the escalation of lung injury observed in critical cases [17,18]. IP-10 is known for its role in recruiting immune cells, particularly T lymphocytes, to sites of infection or inflammation. In COVID-19, increased levels of IP-10 have been observed, suggesting that it may contribute to the recruitment of immune cells to the respiratory system and exaggerate the immune response. Monocytes attracted by MCP-1 can differentiate into macrophages, which play a crucial role in tissue repair and remodelling [19]. MCP-1 may contribute to the inflammatory cascade observed in severe cases [20].

Understanding the intricate roles of these cytokines is essential for unravelling the complexities of severe COVID-19. Targeting specific cytokines may offer potential avenues for therapeutic intervention, aiming to modulate the inflammatory response and individualise treatment in the most severe manifestations of the disease. The timing, duration, and dose of AP administration may have been suboptimal due to a lack of preliminary data on these parameters. In this study, AP was administered shortly after admission to the ICU, as it was expected that this was the moment when patients were likely experiencing a cytokine storm. However, in hindsight, patients requiring oxygen supplementation and/or hospital admission were likely already suffering from an exacerbated immune response. Additionally, AP or placebo was administered for four days, and although there are benefits to the drug’s short half-life, it also means that the drug’s plasma levels become undetectable by day 5. During the COVID-19 pandemic, it was observed that severely ill COVID-19 patients spent an average of 18 days in the hospital and remained hypermetabolic or suffered from a superinfection for an extended period. Therefore, it is possible that the drug was given too late and too short a duration to reveal a clinical benefit. The dose that was used in this study was derived from earlier clinical studies on acute renal failure but may have been suboptimal for our patient population [21]. Nevertheless, given the fact that no safety trial could be conducted for SARS-CoV-2 infected patients, we assumed that this would be a safe dosage. Furthermore, this study ran from December 2020 to March 2022, during which various immunomodulatory therapies and vaccinations were added to standard care. The introduction of new immunomodulatory drugs may have reduced the effect of AP. Also, vaccination appears to have altered the immune response and cytokine storm, leading to a sharp decline in ICU admissions and inclusions for this study [22]. Moreover, the virus mutated multiple times, resulting in differences in the virus’s pathogenicity, leading to heterogeneity among the enrolled patients. This may have resulted in the high variation seen in the cytokine levels of critical cytokines IL6, IP10, and MCP-1, and could possibly have occluded any therapeutic effects. However, subgroup analyses among the patients with high versus low baseline levels of IL-6 and those who did and did not receive tocilizumab did not result in other outcomes. Nevertheless, it is essential to note that the sample sizes in these subgroups were limited, making definitive interpretations impossible.

A limitation of this study was its premature termination due to meeting the threshold for futility, which could indicate insufficient statistical power or unforeseen factors affecting the outcomes. Furthermore, the generalisability may be an issue since this study was conducted within a specific timeframe; therefore, the findings may not be generalisable to patients with COVID-19 admitted to the ICU in different geographical locations or at different stages of the pandemic.

The strengths of the study are its design and safety assessment. This was a randomised controlled trial (RCT) design, which minimised selection bias and allowed for the evaluation of causal relationships between the treatment (AP) and outcomes compared to a placebo. We observed no safety issues associated with AP therapy, indicating that the treatment may be well-tolerated in COVID-19 ICU patients. This finding adds important information to the existing literature on the safety profile of AP.

## 5. Conclusions

In conclusion, we found that AP treatment for severe SARS-CoV-2 patients was safe, but in the current trial set-up, we observed no differences between the alkaline phosphatase and placebo treatments in the need for or duration of mechanical ventilation, nor in the mortality, levels of routine clinical lab parameters, or levels of inflammatory cytokines. However, there remains a possibility that alkaline phosphatase treatment still holds promise as a therapeutic intervention for infections eliciting hyper-inflammatory responses. Additional investigations, incorporating the timing of intervention, optimal dosing, and a longer duration of treatment, are warranted to ascertain the potential impact of alkaline phosphatase as a therapeutic agent for hyper-inflammatory disorders.

## Figures and Tables

**Figure 1 biomedicines-12-00723-f001:**
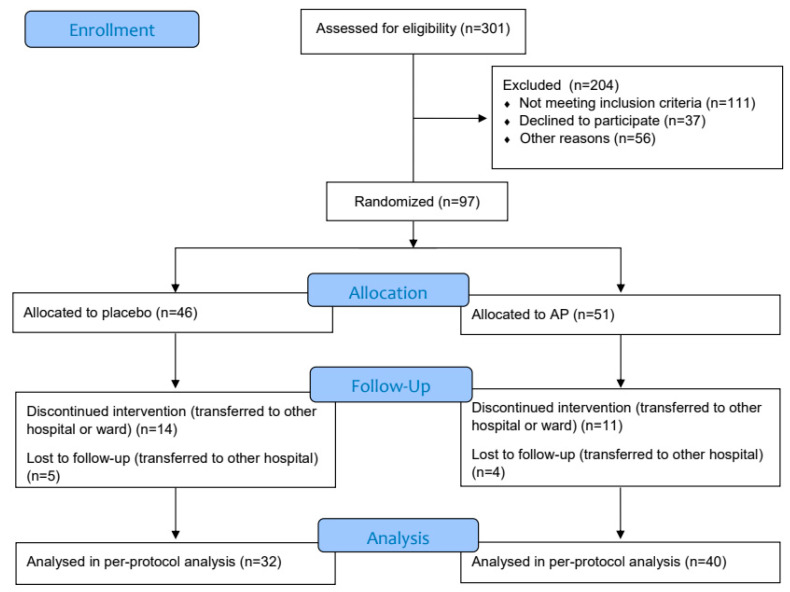
CONSORT flow diagram.

**Figure 2 biomedicines-12-00723-f002:**
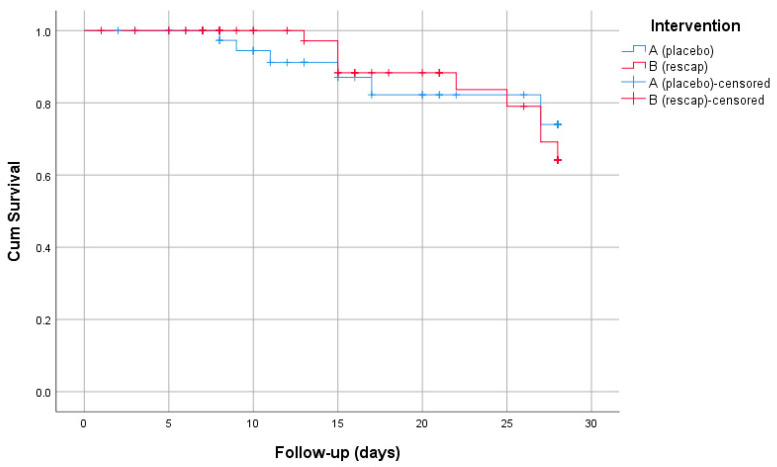
Kaplan–Meier survival curves by treatment group.

**Figure 3 biomedicines-12-00723-f003:**
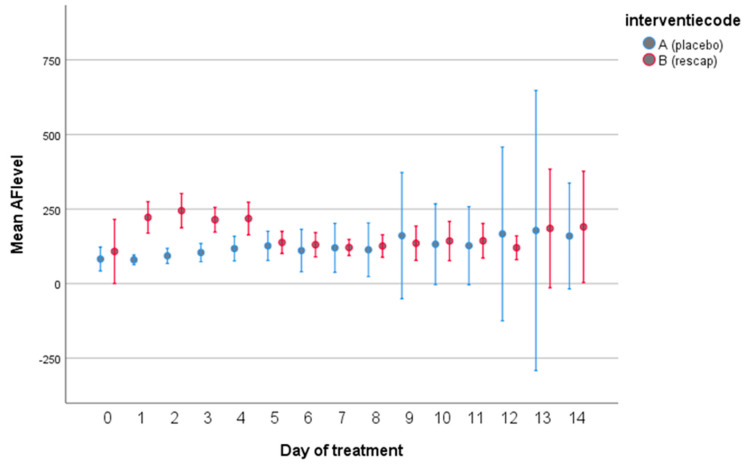
Levels of alkaline phosphatase over 14–day period by treatment group. Based on levels (mean ± 95% confidence intervals) of alkaline phosphatase in 45 patients from Red Cross Hospital, Beverwijk, The Netherlands.

**Figure 4 biomedicines-12-00723-f004:**
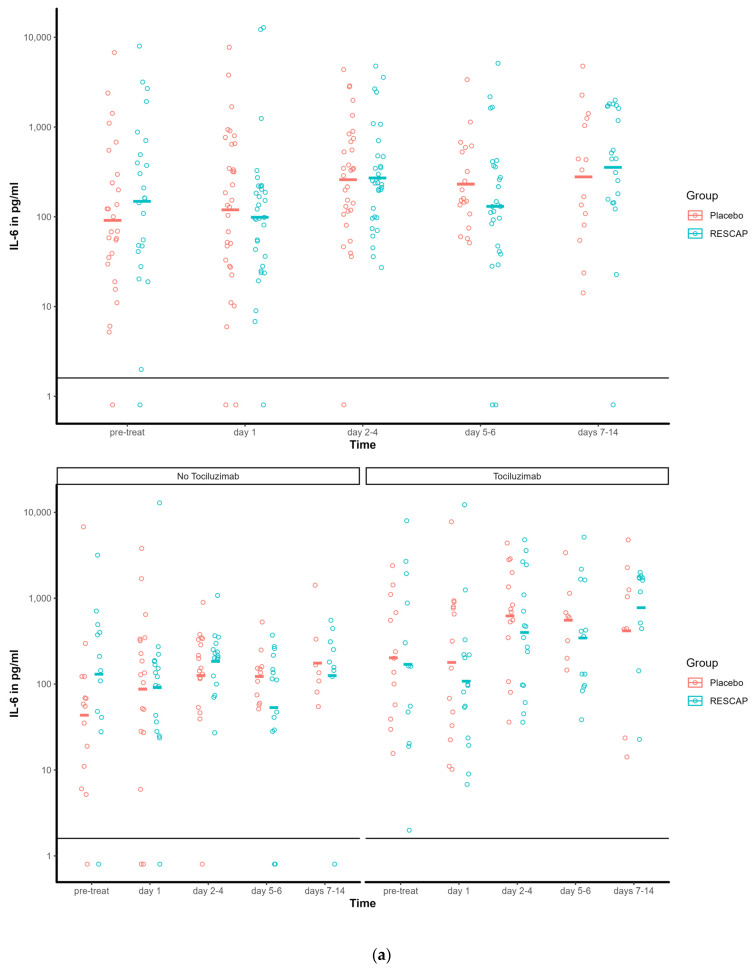
(**a**) Course of values of inflammatory marker IL6 for the entire group and by tocilizumab (no/yes). (**b**) Course of values of inflammatory marker IL10 for the entire group and by tocilizumab (no/yes). (**c**) Course of values of inflammatory marker MCP–1 for the entire group and by tocilizumab (no/yes).

**Table 1 biomedicines-12-00723-t001:** Demographics and baseline characteristics by treatment group.

	Intention to Treat (*N* = 97)			Per Protocol (*N* = 72)		
	Placebo (*n* = 46)	AP (*n* = 51)	*p* Value	Placebo (*n* = 32)	AP (*n* = 40)	*p* Value
Age, yrs (mean, SD)	61.8 (10.0)	60.1 (11.8)	0.453	62.7 (10.2)	61.9 (11.4)	0.752
Sex (*N* male, %)	30 (65%)	39 (77%)	0.222	20 (63%)	30 (75%)	0.253
BMI, kg/m^2^ (mean, SD)	30.0 (4.9)	29.9 (5.4)	0.897	30.5 (4.9)	30.4 (5.9)	0.940
Comorbidities (*N* yes, %)						
Diabetes	9 (20%)	10 (20%)	0.962	7 (22%)	9 (23%)	0.949
Respiratory disease	13 (29%)	5 (10%)	0.017	8 (25%)	5 (13%)	0.171
(COPD/asthma/other)						
Kidney disease	3 (7%)	2 (4%)	0.663	3 (9%)	2 (5%)	0.468
Severe cardiovascular disease	13 (29%)	7 (14%)	0.068	10 (31%)	6 (15%)	0.099
Other ^a^	7 (16%)	5 (10%)	0.395	5 (16%)	5 (13%)	0.703
Vaccinated (*N* yes, %)	12 (26%)	15 (29%)	0.715	11 (34%)	14 (35%)	0.956
PCR (*N* yes, %)	45 (100%)	49 (98%)	-	32 (100%)	39 (98%)	0.368
Time from symptom onset to randomisation, days (median, IQR)	10.0 (4.0) (*n* = 44)	10.0 (3.0) (*n* = 47)	0.122	10.0 (5.0)	9.0 (3.0)	0.975
APACHE IV score on the day of admission to ICU (mean, SD)	44.5 (16.8) (*n* = 43)	46.4 (14.9) (*n* = 51)	0.557	43.9 (15.0)	46.2 (16.2)	0.541
Routine clinical lab values at baseline						
Temperature (med, IQR)	37.3 (1.6)	37.2 (1.4)	0.812	37.3 (1.8)	37.2 (1.1)	0.549
Lowest PaO_2_/FiO_2_ ratio (med, IQR)	97.0 (42.1)	83.0 (56.0)	0.127	98.0 (39.2)	77.5 (45.0)	0.063
CRP (med, IQR) mg/mL	82 (126)	110 (111)	0.120	76 (129)	116 (105)	0.156
HB (Hbm) (med, IQR) mmoL/L	8.5 (1.5)	8.3 (1.2)	0.725	8.4 (1.4)	8.4 (1.4)	0.567
Leukocytes (med, IQR) 10^9^/l	8.6 (4.2)	8.5 (4.7)	0.839	9.1 (4.2)	8.6 (4.9)	0.989
Creatinine (sCream) (med, IQR) μmoL/L	67.0 (23.0)	71.2 (29.0)	0.952	71 (27)	73 (32)	0.858
Dexamethasone (*N* yes, %)	43 (94%)	50 (98%)	0.343	29 (91%)	39 (98%)	0.317
Tociluzumab (*N* yes, %)	14 (30%)	17 (33%)	0.760	9 (28%)	13 (33%)	0.689
Antibiotics (*N* yes, %)	33 (77%)	39 (78%)	0.885	22 (73%)	29 (74%)	0.923
Antibiotics, *N* days (med, IQR)	5 (3)	5 (17)	0.264	7 (7)	7 (17)	0.244

SD, standard deviation; med, median; IQR, interquartile range. ^a^ Other comorbidities include: alcohol abuse, morbid obesity, hypertension, heart rhythm disorder, depression, hypothyroidy, CVA in history, and latent TBC.

**Table 2 biomedicines-12-00723-t002:** Clinical outcomes by treatment group.

	Intention to Treat (*N* = 97) *			Per Protocol (*N* = 72)		
	Placebo (*n* = 46)	AP (*n* = 51)	*p* Value	Placebo (*n* = 32)	AP (*n* = 40)	*p* Value
Mechanical ventilation (*N* yes, %)	26 (57%)	35 (69%)	0.218	17 (53%)	29 (73%)	0.089
Duration of mechanical ventilation (days; no = 0 days) (median, IQR)	3.5 (11.8)	6.1 (15.0)	0.102	3.5 (15.6)	7.3 (24.1)	0.183
Duration of mechanical ventilation (days) among patients receiving mechanical ventilation (median, IQR)	9.3 (17.4)	9.0 (21.2)	1.000	14.8 (20.4)	12.0 (22.0)	0.841
Mortality (*N*, %)	8 (17%)	11 (22%)	0.605	6 (19%)	10 (25%)	0.526
Mortality at 28 days (*N*, %)	6 (13%)	9 (18%)	0.531	4 (13%)	8 (20%)	0.396
Length of ICU stay, days (median, IQR)	12 (10)	11 (20)	1.000	13 (14)	16 (22)	0.663
Length of ICU stay excluding 9 patients lost to follow-up, days (median, IQR)	12 (10)	11 (21)	0.796	13 (15)	15 (24)	0.379
Length of hospital stay, days (median, IQR)	18 (15)	19 (22)	0.398	19 (20)	23 (23)	0.360

AP, alkaline phosphatase; IQR, interquartile range * Analyses include data from 9 patients who were lost to follow-up within the 28-day study period, i.e., were transferred to an ICU in a non-participating hospital, after which no information on clinical outcomes could be obtained.

**Table 3 biomedicines-12-00723-t003:** Effects of AP versus placebo on 14-day course of routine clinical lab parameters.

	Intention to Treat (*N* = 97)			Per Protocol (*N* = 72)		
	Estimate ^1^	SE	*p* Value	Estimate ^1^	SE	*p* Value
Temperature	0.1	0.1	0.444	0.1	0.1	0.707
Lowest PaO_2_/FiO_2_ ratio	5.2	6.9	0.457	8.6	7.7	0.269
CRP mg/L	2.3	11.0	0.838	7.7	12.3	0.531
Hbm mmol/L	−0.2	0.2	0.254	−0.2	0.2	0.480
Leukocytes 10^9^/L	0.7	0.6	0.270	0.8	0.7	0.253
Creatinine μmoL/L	3.7	7.0	0.600	5.2	8.8	0.561

^1^ Estimate of the overall effect of the AP treatment versus the placebo over the course of 14 days, within patients and between patients; a positive (negative) value indicates that AP exerted an x-unit higher (lower) level in the parameter under investigation than the placebo over the course of 14 days.

**Table 4 biomedicines-12-00723-t004:** Effect of AP versus placebo on 14-day course of inflammatory markers (cytokines).

	Intention to Treat (*N* = 65) 31 Placebo, 34 AP			Per Protocol (*N* = 44) 20 Placebo, 24 AP		
	Estimate ^1^	SE	*p* Value	Estimate ^1^	SE	*p* Value
Anti-inflammatory						
IL10	3.3	5.7	0.561	−1.8	1.8	0.323
TGFb1	ND	ND	ND	ND	ND	ND
Pro-inflammatory						
IL1b	−0.9	5.6	0.871	−1.2	7.0	0.858
IL6	-26.5	210.1	0.900	161.3	243.4	0.511
IL8	7.1	9.5	0.458	5.7	10.6	0.593
IL12p70	−0.3	1.2	0.832	0.9	1.4	0.495
IL17a	−0.8	1.5	0.574	−1.3	1.6	0.934
IP10	−3.9	58.8	0.948	24.4	69.3	0.727
MCP1	8.8	72.7	0.903	57.4	90.6	0.529
TNFa	ND	ND	ND	ND	ND	ND
IFNg	ND	ND	ND	ND	ND	ND
Adaptive						
IL4	−7.3	10.3	0.479	−6.6	12.3	0.590
IL2	ND	ND	ND	ND	ND	ND
Ratio						
IL6/IL10	−5.3	20.2	0.793	4.5	23.2	0.846

SE, standard error; ND, not detectable. For these cytokines, more than 50% of the daily values were below the lowest detection limit and were, therefore, classified as not statistically analysable. ^1^ Estimate of the overall effect of the AP treatment versus the placebo over the course of 14 days, within patients and between patients; a positive (negative) value indicates that AP exerted a x-unit higher (lower) level in the parameter under investigation than the placebo over the course of 14 days.

## Data Availability

The dataset underlying the conclusions of our article is available following contact to the corresponding author. Restrictions for data sharing may apply, for example, depending on the aim of the data request and the scope of the informed consent provided by the participants and the EU General Data Protection Regulation.

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
