# Peer review of "Efficacy of Alkaline Phosphatase in Critically Ill Patients with COVID-19: A Multicentre Investigator-Initiated Double-Blind Randomised Placebo-Controlled Trial"

_biomedicines, 2024, doi:10.3390/biomedicines12040723_

Round 1

Reviewer 1 Report

Comments and Suggestions for Authors

Congratulations for your efforts to investigate novel anti-inflammatory compounds that might be proved beneficial for the SARS-CoV-2 patients. The lack of knowledge and and the shortage of supportive medication support the quest for modern antiviral weapons. However I have only 2 comments in order to assist the authors to present more efficiently the study procedure and results.

Comment 1:  Please explain further about your decision for the route and the dosage of alkaline phosphatase you administered to the patients.

Comment 2: Please state limitations and possible strengths of your study

Author Response

Response to Reviewer 1

Congratulations for your efforts to investigate novel anti-inflammatory compounds that might be proved beneficial for the SARS-CoV-2 patients. The lack of knowledge and and the shortage of supportive medication support the quest for modern antiviral weapons. However I have only 2 comments in order to assist the authors to present more efficiently the study procedure and results.

Thank you very much for taking the time to review this manuscript. Please find the detailed responses below and the corrections highlighted in the re-submitted files.

Comment 1:  Please explain further about your decision for the route and the dosage of alkaline phosphatase you administered to the patients.

We added the following sentence to the manuscript (lines 123-125): The chosen dose regime is retrieved from modelling studies of kinetics data from two clinical studies with the same Alkaline Phosphatase (AP) and the same total dose of 10,000 units of which 1000 units were given as a bolus.[16]

  1. Presbitero A, Mancini E, Brands R, Krzhizhanovskaya VV, Sloot PMA. Supplemented Alkaline Phosphatase Supports the Immune Response in Patients Undergoing Cardiac Surgery: Clinical and Computational Evidence. Front Immunol. 2018 Oct 11;9:2342. doi: 10.3389/fimmu.2018.02342. PMID: 30364262; PMCID: PMC6193081.

Comment 2: Please state limitations and possible strengths of your study

Thank you for pointing this out. We have added the following two paragraph to the discussion section (lines 345-356):

A limitation of the study was the prematurely termination due to meeting the threshold for futility, which could indicate insufficient statistical power or unforeseen factors affecting the outcomes. Furthermore the generalizability may be an issue since this study was conducted within a specific timeframe; therefore the findings may not be generalizable to patients with COVID-19 admitted to the ICU in different geographical locations or at different stages of the pandemic.

The strengths of the study are the design and safety assessment. This is a Randomized Controlled Trial (RCT) design and minimizes selection bias and allows for the evaluation of causal relationships between treatment (AP) and outcomes compared to placebo. We observed no safety issues associated with AP therapy, indicating that the treatment may be well-tolerated in COVID-19 ICU patients. This finding adds important information to the existing literature on the safety profile of AP.

Reviewer 2 Report

Comments and Suggestions for Authors

Manuscript titled "Efficacy of Alkaline Phosphatase on duration of mechanical ventilation and immunopathology in critically ill patients with COVID-19: a multicentre investigator-initiated double-blind randomized placebo-controlled trial" is a nice piece of work. Although AP do not have any positive outcome in terms of treatment but study established as safe nature of the enzyme. The over all draft looks great. I do not have any specific reservation against manuscript and should be accepted after minor revision. My comments to authors are below.

1) Make title concise, it is too lengthy.

2) Will be good to discuss good and bad aspects of this pandemic in introduction or include relevant literature (see https://www.mdpi.com/2673-8112/3/12/121) .

3) Author should explain possible reason for observed non effectiveness of AP in this trail.

4) Good to include protein or molecules which sowed positive effect on COVID19 treatment.

5) Information related to strains which infected the individuals included in study should be included in draft

6) Minor writing improvement is needed

Comments on the Quality of English Language

Overall writing is OK

Author Response

Response to Reviewer 2:

Manuscript titled "Efficacy of Alkaline Phosphatase on duration of mechanical ventilation and immunopathology in critically ill patients with COVID-19: a multicentre investigator-initiated double-blind randomized placebo-controlled trial" is a nice piece of work. Although AP do not have any positive outcome in terms of treatment but study established as safe nature of the enzyme. The over all draft looks great. I do not have any specific reservation against manuscript and should be accepted after minor revision. My comments to authors are below.

Thank you very much for taking the time to review this manuscript. Please find the detailed responses below and the corrections highlighted in the re-submitted files.

1) Make title concise, it is too lengthy.

We reduced the number of words in the title which is now: Efficacy of Alkaline Phosphatase on duration of mechanical ventilation and immunopathology in critically ill patients with COVID-19: a multicentre investigator-initiated double-blind randomized placebo-controlled trial"

2) Will be good to discuss good and bad aspects of this pandemic in introduction or include relevant literature (see https://www.mdpi.com/2673-8112/3/12/121) .

We added/rephrased the following sentence in the introduction: COVID-19 caused by the coronavirus Severe Acute Respiratory Syndrome Coronavirus 2 (SARS-CoV-2) lead to an unprecedented global burden, and simultaneously it has acted as an catalyst for new medical therapies and insights.

3) Author should explain possible reason for observed non effectiveness of AP in this trail.
& 4) Good to include protein or molecules which sowed positive effect on COVID19 treatment.

The timing, duration, and dose of AP administration may have been suboptimal. In the discussion we have stated some ideas about the lack of efficiency. We question timing, duration and dose of the medication. Also we discuss the introduction of other immunomodulatory drugs and the emergence of different strains of the virus. We also mentioned other immunomodulatory drugs. See lines 297-345.

5) Information related to strains which infected the individuals included in study should be included in draft

We understand this comment, however, unfortunately, we don’t have data on which strains infected the individuals in this study.

6) Minor writing improvement is needed

We reviewed our paper ourselves and improved some phrases and typo’s.

Reviewer 3 Report

Comments and Suggestions for Authors

The manuscript refers to a clinical trial analysing alkaline phosphatase's possible role in mechanical and immunopathology's duration in critically ill patients with SARS-CoV-2 infection. Even though the hypothesis behind the clinical trial does not seem adequate, the trial's design, the follow-up and the analysis of different patterns were performed adequately. The statistical analysis is adequate, and the conclusions based on the results are reasonable. However, some minor issues may enhance the quality of the manuscript. How many patients were hypertensive? Was dexamethasone used as therapy in some patients? If so, are there any differences? The paraclinical analysis of coagulation factors, D dimer, and PF4 must be included in addition to the number of platelets, reference  10.1371/journal.pone.0287117

Please add in Table 3 the lymphocyte/neutrophil ratio, which is considered an important parameter for severity rather than total leukocytes. Please correct Table 4 IL1 beta is pro-inflammatory and change the abbreviations. NA usually refer to non analysed and is confusing if no cytokine is detectable using ND. Please modify the label of the figure. It gets confusing for each cytokine. Looking it closely, it would be very interesting to do a ratio of IL6/IL10 in treated and not-treated individuals since there seems to be a trend which may be a valuable index, as suggested by some authors. 10.1016/j.heliyon.2023.e16985

Finally, the authors have to add, as a section, the limitations of the study, which range from several factors. 

Author Response

Response to Reviewer 3

The manuscript refers to a clinical trial analysing alkaline phosphatase's possible role in mechanical and immunopathology's duration in critically ill patients with SARS-CoV-2 infection. Even though the hypothesis behind the clinical trial does not seem adequate, the trial's design, the follow-up and the analysis of different patterns were performed adequately. The statistical analysis is adequate, and the conclusions based on the results are reasonable. However, some minor issues may enhance the quality of the manuscript.

Thank you very much for taking the time to review this manuscript. Please find the detailed responses below and the corrections highlighted in the re-submitted files.

How many patients were hypertensive?

Only severe cardiovascular co-morbidity was scored.

Was dexamethasone used as therapy in some patients? If so, are there any differences?

Yes, most patients received dexamethasone; only 4 did not. We added this information to Table 1. There was no difference between de AP and placebo group.

The paraclinical analysis of coagulation factors, D dimer, and PF4 must be included in addition to the number of platelets, reference 10.1371/journal.pone.0287117

We understand this comment, however, unfortunately, only platelets were routinely monitored.

Please add in Table 3 the lymphocyte/neutrophil ratio, which is considered an important parameter for severity rather than total leukocytes.

We understand this comment, however, unfortunately, leukocyte differentiation was not routine measured. CRP and PCT were routinely monitored.

Please correct Table 4 IL1 beta is pro-inflammatory and change the abbreviations. NA usually refer to non analysed and is confusing if no cytokine is detectable using ND. Please modify the label of the figure. It gets confusing for each cytokine. Looking it closely, it would be very interesting to do a ratio of IL6/IL10 in treated and not-treated individuals since there seems to be a trend which may be a valuable index, as suggested by some authors. 10.1016/j.heliyon.2023.e16985.

Il1beta was moved to the pro-inflammatory category. We replaced NA by ND. We added the ratio IL6/IL10 to Table 4; there was not difference between the AP and placebo group.

Finally, the authors have to add, as a section, the limitations of the study, which range from several factors.

Thank you for pointing this out. We have added the following paragraph to the discussion section (lines 345 and further:

A limitation of the study was the prematurely termination due to meeting the threshold for futility, which could indicate insufficient statistical power or unforeseen factors affecting the outcomes. Furthermore the generalizability may be an issue since this study was conducted within a specific timeframe; therefore the findings may not be generalizable to patients with COVID-19 admitted to the ICU in different geographical locations or at different stages of the pandemic.